META-RESEARCH ARTICLE

# Analysis of animal-to-human translation shows that only 5% of animal-tested therapeutic interventions obtain regulatory approval for human applications

**Benjamin V. Ineichen** [1,2]*, **Eva Furrer**[1], **Servan L. Grüninger**[1,3], **Wolfgang E. Zürrer**[1], **Malcolm R. Macleod**[4]

**1** Centre for Reproducible Science, University of Zurich, Zurich, Switzerland, **2** Clinical Neuroscience Center, University of Zurich, Zurich, Switzerland, **3** Department of Mathematics, University of Zurich, Zurich, Switzerland, **4** Centre for Clinical Brain Sciences, The University of Edinburgh, Edinburgh, United Kingdom

* benjamin.ineichen@uzh.ch

**Data Availability Statement:** All data and code that support the findings of this study are available at

## Abstract

There is an ongoing debate about the value of animal experiments to inform medical practice, yet there are limited data on how well therapies developed in animal studies translate to humans. We aimed to assess 2 measures of translation across various biomedical fields: (1) The proportion of therapies which transition from animal studies to human application, including involved timeframes; and (2) the consistency between animal and human study results. Thus, we conducted an umbrella review, including English systematic reviews that evaluated the translation of therapies from animals to humans. Medline, Embase, and Web of Science Core Collection were searched from inception until August 1, 2023. We assessed the proportion of therapeutic interventions advancing to any human study, a randomized controlled trial (RCT), and regulatory approval. We meta-analyzed the concordance between animal and human studies. The risk of bias was probed using a 10-item checklist for systematic reviews. We included 122 articles, describing 54 distinct human diseases and 367 therapeutic interventions. Neurological diseases were the focus of 32% of reviews. The overall proportion of therapies progressing from animal studies was 50% to human studies, 40% to RCTs, and 5% to regulatory approval. Notably, our meta-analysis showed an 86% concordance between positive results in animal and clinical studies. The median transition times from animal studies were 5, 7, and 10 years to reach any human study, an RCT, and regulatory approval, respectively. We conclude that, contrary to widespread assertions, the rate of successful animal-to-human translation may be higher than previously reported. Nonetheless, the low rate of final approval indicates potential deficiencies in the design of both animal studies and early clinical trials. To ameliorate the efficacy of translating therapies from bench to bedside, we advocate for enhanced study design robustness and the reinforcement of generalizability.

https://osf.io/frjm4 (data including meta-data) and https://osf.io/9fgru (R code).

**Funding:** Swiss National Science Foundation (No. 407940_206504, to BVI) UZH Digital Entrepreneur Fellowship (No number, to BVI). UFAW (Universities Federation for Animal Welfare, to BVI). The sponsors had no role in the design and conduct of the study; collection, management, analysis, and interpretation of the data; preparation, review, or approval of the manuscript; and decision to submit the manuscript for publication.

**Competing interests:** The authors have declared that no competing interests exist.

## Introduction

Animal studies remain foundational in basic research, accounting for a substantial share of global biomedical research investment. These experiments have provided insight on aspects of human diseases and have paved the way for therapeutic innovations. For example, mitoxantrone and glatiramer acetate, FDA-approved drugs for multiple sclerosis, owe their inception at least partly to animal studies [1]. Yet, in recent years, concerns have grown about the low translatability of findings from animal experiments to humans, a concern that certain drugs with beneficial findings in animal experiments did not show similar effects in humans [2–4]. For example, while NXY-059 showed substantial promise in animal stroke studies, it failed in human trials [5]. Natalizumab displayed considerable efficacy in both animal and human multiple sclerosis trials, but the animal studies did not detect a severe side effect caused by a virus not present in rodents [6,7]. Opicinumab demonstrated significant promise in multiple sclerosis animal studies but failed its primary endpoint in human trials for multiple sclerosis [8,9].

The concerns of low translatability of animal research are particularly relevant to the debate on the ethical use of animals in research because clinical translation is one of the primary justifications for such research [10]. Discussions around the usefulness of animal experiments persist, but much of the current debate relies on anecdotal findings from discrete research areas [11]. High-level evidence—spanning various biomedical sectors and assessing translational success rates—is scarce. Hence, here we aimed to (1) offer a quantitative perspective on animal-to-human translation across diverse biomedical fields; (2) scrutinize the agreement between findings in animal and human drug development studies; and (3) probe the time intervals separating animal and human trials during drug development.

## Materials and methods

### Study registration

We registered the study protocol on the Open Science Framework platform (OSF, https://osf.io/jh2d8) and used the Preferred Reporting Items for Systematic Reviews and Meta-Analysis (PRISMA) guidelines for reporting [12].

### Approach to identification of therapeutic interventions

Our method to identify therapeutic interventions assessed for bench-to-bedside translation was identified via a two-stage process: in a first step, we identified systematic/scoping reviews assessing animal-to-human translation of specific therapeutic interventions (described in the Methods section "Search strategy for systematic reviews"). In the second step, these therapeutic interventions were included in our analysis of translational proportions, consistency of animal and human findings, and development times (described in the Methods section "Assessment of translational proportions and development times").

### Search strategy for systematic reviews

We searched for studies published from inception up to August 01, 2023, in Medline (Ovid), Embase, and Web of Science Core Collection (Clarivate). We created the search string in Medline and translated to the other databases. The exact search strings are provided in the S1 Data. In brief, the search string comprised one block with terms for translation and one block for systematic/scoping reviews, and was limited to animal studies by employing the SYRCLE animal filter [13]. To probe the sensitivity of this approach, we also tested a broad search string comprising only the systematic review block and the SYRCLE animal filter (i.e., without the block for translation). Our protocol stated that we would use this broader strategy if we

identified in a subgroup of search returns additional studies equivalent to 5% of the total. However, we did not identify any additional eligible studies using the broad search strategy (i.e., 0%) and so we used the narrower search string for this systematic review (see protocol for details). We deduplicated the references in Endnote using the Bramer method [14].

## Inclusion and exclusion criteria

**Inclusion criteria.** Systematic reviews or scoping reviews with or without meta-analysis which investigate translation of interventions in animal models of human diseases, i.e., the study must have the goal of assessing animal-to-human translation of therapies. Any type of intervention with the goal of improving at least 1 disease outcome was eligible (e.g., drugs, surgical interventions, neuromodulation, diets, behavioral therapy). The minimum requirement to be eligible as systematic review, scoping review, and/or meta-analysis was at abstract level: (1) Having at least 2 authors; (2) mentioning a systematic literature search; and (3) at full-text level having a paper section describing methodology of the systematic review.

**Exclusion criteria.** Original studies and/or studies not assessing bench-to-bedside translation, non-English articles, and gray literature (conference abstracts, book chapters). We excluded non-systematic reviews but retained them to find potential additional references.

## Study selection and data extraction

Three independent reviewers (BVI, SG, and EF) screened titles and abstracts of studies for their relevance in the web-based application SyRF (RRID: SCR_018907) in duplicate [15]. We resolved discrepancies by discussion. Subsequently to full-text screening, we extracted the following data: bibliographic data (author names, journal, title, publication year, digital object identifier), number of included animal studies and clinical trials, disease classes, any data related to translation, any provided definition on translation as well as data on year/outcome of clinical studies (see below). We extracted all data from text/tables if possible, and if not extracted from figures using Universal Desktop Ruler [16].

## Critical appraisal of included studies

We assessed the quality of each included study against predefined criteria by 3 independent reviewers in duplicate (BVI, WEZ, and EF), based on a checklist proposed by Sena and colleagues [17], and extended with additional items for a more granular critical appraisal. Concretely: (1) Was an a priori study protocol defined? (2) Was a flowchart for study selection provided? (3) Was a conflict-of-interest statement provided? (4) Was screening and/or extraction conducted by 2 or more reviewers? (5) Was a clear research question defined? (6) Were in- and exclusion criteria reported? (7) Were 2 or more literature databases searched? (8) Was a search date provided? (9) Was a search string provided? (10) Was a critical appraisal conducted? (11) Did the study mention alignment with relevant guidelines, e.g., SYRCLE, CAMARADES, or PRISMA? We resolved discrepancies by discussion. This appraisal method was initially tested in a separate umbrella review. Detailed application guidelines are available in the S2 Data. The inter-rater agreement was calculated using Cohen's Kappa. Of note, we included all systematic reviews into our final analysis, regardless of their risk of bias.

## Assessment of translational proportions and development times

**Definition of translation.** We used the following working definition of translation: the process of turning observations from animal experiments into interventions that improve the health of human individuals and the public [18].

**Study year and outcomes.** For each individual intervention identified in the eligible systematic reviews, 2 independent reviewers extracted: (1) the first published animal study testing the respective intervention; and (2) the first clinical study testing the respective intervention. We included any type of clinical studies including pilot studies or case series, and (3) the first randomized controlled trial (RCT) testing the respective intervention. In addition, for all clinical studies/RCTs, we extracted the main study outcome as defined by the authors of the respective study, grouped into 4 classes, i.e., whether the intervention had a positive, negative, mixed, or neutral effect on the corresponding disease outcome [19], any outcome was considered, e.g., primary or secondary outcome. We extracted these data in first priority from the respective systematic reviews. If these data were not available in the systematic reviews, we searched Medline (Ovid) and Embase for corresponding clinical studies/RCTs. For this, we used a search string comprising the intervention and disease name, including synonyms. For clinical approval of an intervention, we considered FDA approval (by searching the FDA webpage for respective therapies) as well as UK and Swiss medical guidelines for recommended use of respective interventions. We resolved discrepancies by discussion.

## Data synthesis and analysis

**Narrative synthesis and descriptive statistics.** We provide a narrative summary of the extent of bench-to-bedside translation, supported by descriptive statistics of extracted parameters.

**Meta-analysis on relative risks.** To assess the concordance between animal and human studies, we conducted a meta-analysis on relative risks, i.e., the ratio of the proportion of positive animal studies to the proportion of positive clinical studies. We conducted this only in case of specific interventions for specific diseases entities (e.g., atorvastatin for glioblastoma). To reduce noise of the dataset, we restricted our analysis to therapies that were the subject of 5 or more published animal studies. As primary outcome, we pooled relative risks to obtain an overall relative risk and 95% confidence intervals. We fitted a random-effects model to the data [20] and estimated the amount of heterogeneity, i.e., $\tau^2$, using the DerSimonian—Laird estimator. We calculated the Q-test for heterogeneity and the $I^2$ statistic. We used the R package *meta* for the meta-analysis [21] (RRID: SCR_019055).

**Kaplan–Meier survival analysis for development times.** To estimate the time-to-event from first animal study to clinical studies/RCTs and eventually official endorsement (i.e., the lag time), we conducted a Kaplan—Meier analysis. We used the packages *survminer* (RRID: SCR_021094) and *survival* (RRID: SCR_021137) for this survival analysis.

All statistical analyses were conducted in the R programming environment (version 4.2.2). We considered a two-tailed $P$ value < 0.05 statistically significant.

## Results

### Eligible publications and general study characteristics

**Eligible studies.** In total, 5,227 original publications were retrieved from our database search, and 1 publication from reference lists of reviews on related topics. After abstract and title screening, 656 publications were eligible for full-text search. After screening the full text of these studies, 122 articles (2% of deduplicated references) were included for qualitative synthesis (Table 1) and a subset of 62 for quantitative synthesis (Fig 1).

The eligible systematic reviews comprised a total of 4,443 animal studies and 1,516 clinical studies (median 21 animal and 8 human studies per systematic review).

Most studies have been published in the last 5 years (88 since 2018). Most studies were from the United States of America (27 studies, 22%), Canada (19, 16%), the Netherlands, (16,

**Table 1. Number of included translational systematic reviews, diseases/conditions, and therapeutic interventions.**

| | | | |
|---|---|---|---|
| Number of systematic reviews included | | 122 | |
| Number of unique diseases/conditions | | 54 | |
| Number of unique therapeutic interventions (including drug and non-drug interventions) | 367 | No. in any human study | 165 (45%) |
| | | No. in an RCT | 132 (36%) |
| | | No. with (FDA) approval | 14 (4%) |

The percentage in brackets refers to the relative numbers of therapeutic interventions entering the respective therapy development stage.

The data underlying this table can be found on https://osf.io/frjm4 (Sheet: *Mastersheet_including_RoB*).

FDA, food and drug administration; No., number; RCT, randomized controlled trial.

13%), Australia (13, 11%), Italy (13, 11%), and the United Kingdom (10, 8%). Thirty-six studies were collaborative efforts between 2 or more countries.

**Diseases and therapies.** The studies covered 54 unique different human diseases/conditions (Table 1). The most common ICD-11 disease classes addressed by the systematic reviews

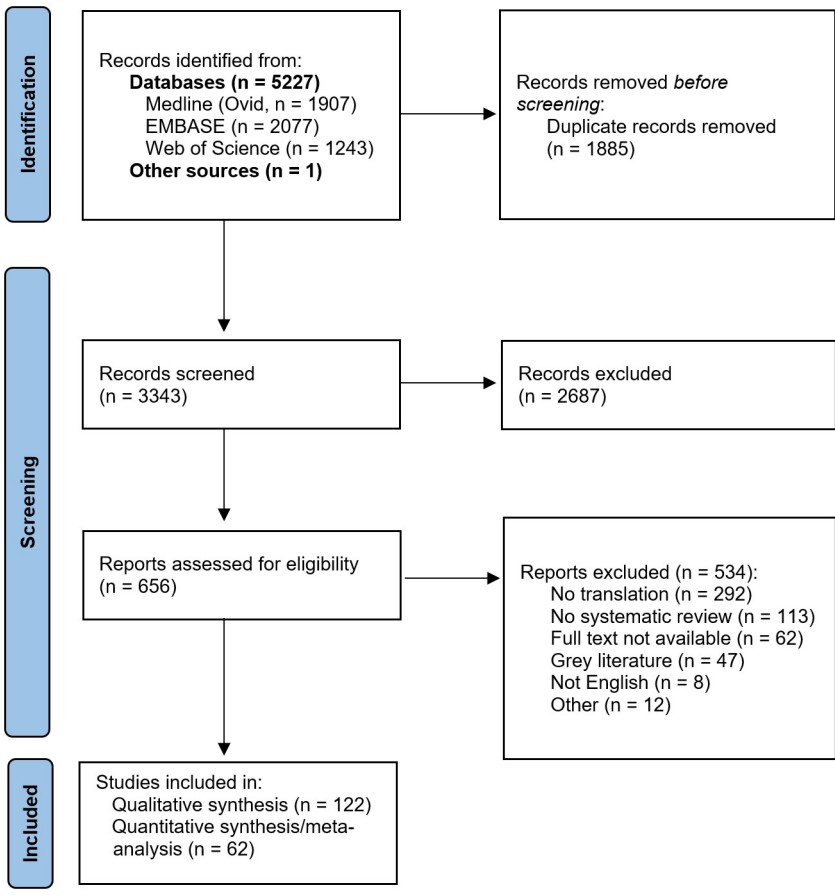

**Fig 1. Flow chart for study inclusion.** Flow chart for inclusion of systematic reviews or scoping reviews with or without meta-analysis which investigate translation of interventions in animal models of human diseases.

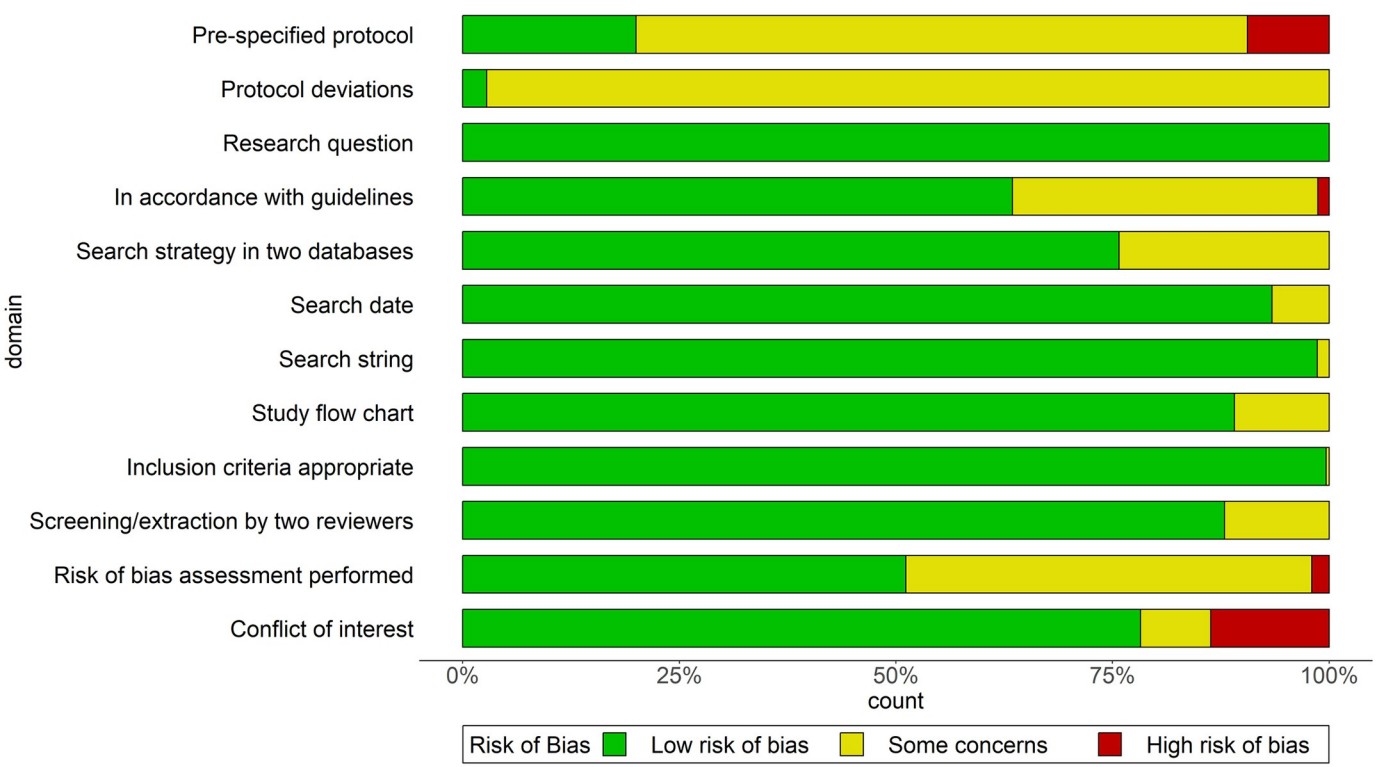

**Fig 2. Risk of bias assessment for included systematic reviews.** The data underlying this figure can be found on https://osf.io/frjm4 (Sheet: *Mastersheet_including_RoB*). The code underlying this figure can be found S1 Code (https://osf.io/9fgru).

were "diseases of the nervous system" (32% of studies), "diseases of the musculoskeletal system and connective tissue" (11%), "mental and behavioral disorders" (9%), "diseases of the circulatory system" (9%), "diseases of the digestive system" (8%), and "neoplasms" (8%).

These reviews included a total of 367 unique therapeutic interventions including drug and non-drug treatments (Table 1). The median interval between the first animal experiment and publication of the respective systematic review was 15 years (range: 3 to 63 years, total observation time 6,736 years). All therapies and diseases are listed in S1 Table.

**Risk of bias assessment.** The inter-rater agreement Kappa was 0.76. We considered most of the included reviews to be at low risk of bias for providing a search date, a search string, a study flow chart, and reporting screening and data extraction by 2 reviewers. However, few studies published a study protocol, and a substantial number of systematic reviews did not perform a risk of bias assessment, thus posing high risk of bias in these domains (Fig 2). Given that most included systematic reviews were deemed at low risk of bias, we anticipated no relevant impact on our overall conclusions.

## Qualitative summary of translation in different biomedical fields

Therapies were tested in a variety of diseases, including neurological, musculoskeletal, psychiatric, circulatory system, digestive, skin, lung, and metabolic diseases (S2–S8 Tables).

Most studies discussed potential hurdles for translation of findings from animal to human studies. A prevalent observation was the disparity in methodological approaches between animal experiments and human studies. Specifically, several studies highlighted that experimental conditions in animal research often do not mirror clinically relevant scenarios [22–33]. For

example, treatments for stroke were frequently tested on young, healthy animals, which contrast the typically multimorbid elderly patient population in clinical settings [22]. Poor study quality and inadequate reporting, predominantly in animal studies, were also recurrent concerns [34–36]. A noticeable reduction in effect size from animal to human studies was documented by several reviews [24,37–40]. This trend was substantiated by a systematic review and meta-analysis which examined the preclinical-to-clinical development trajectory of 37 treatments for acute ischemic stroke, encompassing 50 phase 3 clinical trials, 75 early clinical trials, and 209 animal studies [36]: It observed a progressive reduction in efficacy from animal research to early clinical trials and then to Phase 3 clinical trials. This decline was attributed to shortcomings in preclinical study rigor (such as the absence of randomization and blinding), differences in study design, including the use of differing outcomes, and insufficient statistical power in both animal studies and preliminary clinical studies.

Another comprehensive systematic review covering therapies for cardiac arrest encompassed 415 animal and 43 clinical studies. This review, which evaluated 190 pharmacological interventions, found a limited number of positive outcomes in clinical studies [41]. In addition, many animal studies were conducted subsequent to the publication of a corresponding clinical study [41]. And similar to stroke, there were substantial variation in experimental methodologies between animal and human studies. For example, drugs were typically administered approximately 9.5 min post-cardiac arrest in animal studies, compared to roughly 19.4 min in human studies [42].

Finally, one study traced the developmental pathway of an oncolytic virus used in cancer therapy [43]. While animal experiments exhibited 80% to 100% regression rates in tumours, human studies only demonstrated a range of 0% to 24%. Intriguingly, more rigorously conducted studies showed smaller effect sizes. And the authors emphasize that even successful biotherapeutic interventions might not present a straight-forward translational journey.

## Quantitative overview of therapy translation overall and by discipline

In these systematic reviews, and only accounting for therapies where more than 10 years has elapsed since the initial animal experiment, 50%, 40%, and 5% of therapies entered any human study, an RCT, or have been (FDA) approved (281 therapies, Fig 3A). Diseases of the circulatory system (166 therapies, including stroke: 34%, 29%, and 1%, respectively) and mental health disorders (16 therapies: 50%, 31%, and 0%, respectively) showed particularly low translational proportions (Fig 3B and 3D). Diseases of the musculoskeletal system (13 therapies, 100%, 62%, and 15%, respectively) and cancer (15 therapies, 73%, 47%, and 20%, respectively) showed relatively high translational proportions (Fig 3E and 3F). Translational proportions considering all therapies independent of time gap between animal and clinical studies were slightly lower (Table 1 and S1 Fig). Translational proportions per disease are illustrated in Fig 4 with cardiac arrest, multiple sclerosis, and stroke assessing the most therapeutic interventions in animals (S9 Table).

## Chronology of animal-to-human translation

Because less than half of the therapies transitioned from animal experiments to clinical studies, we were not able to estimate the overall median durations (Fig 5). However, if only considering therapeutic interventions entering any clinical study, an RCT, or obtaining (FDA) approval, the median lag times were 5 years [95% CI: 5 to 6], 7 years [95% CI: 6 to 8], and 10 years [95% CI: 4-not estimable], respectively (S2 Fig). The maximum time from first animal study to any clinical trial, and RCT, or FDA approval was 44 years, 58 years, and 34 years, respectively.

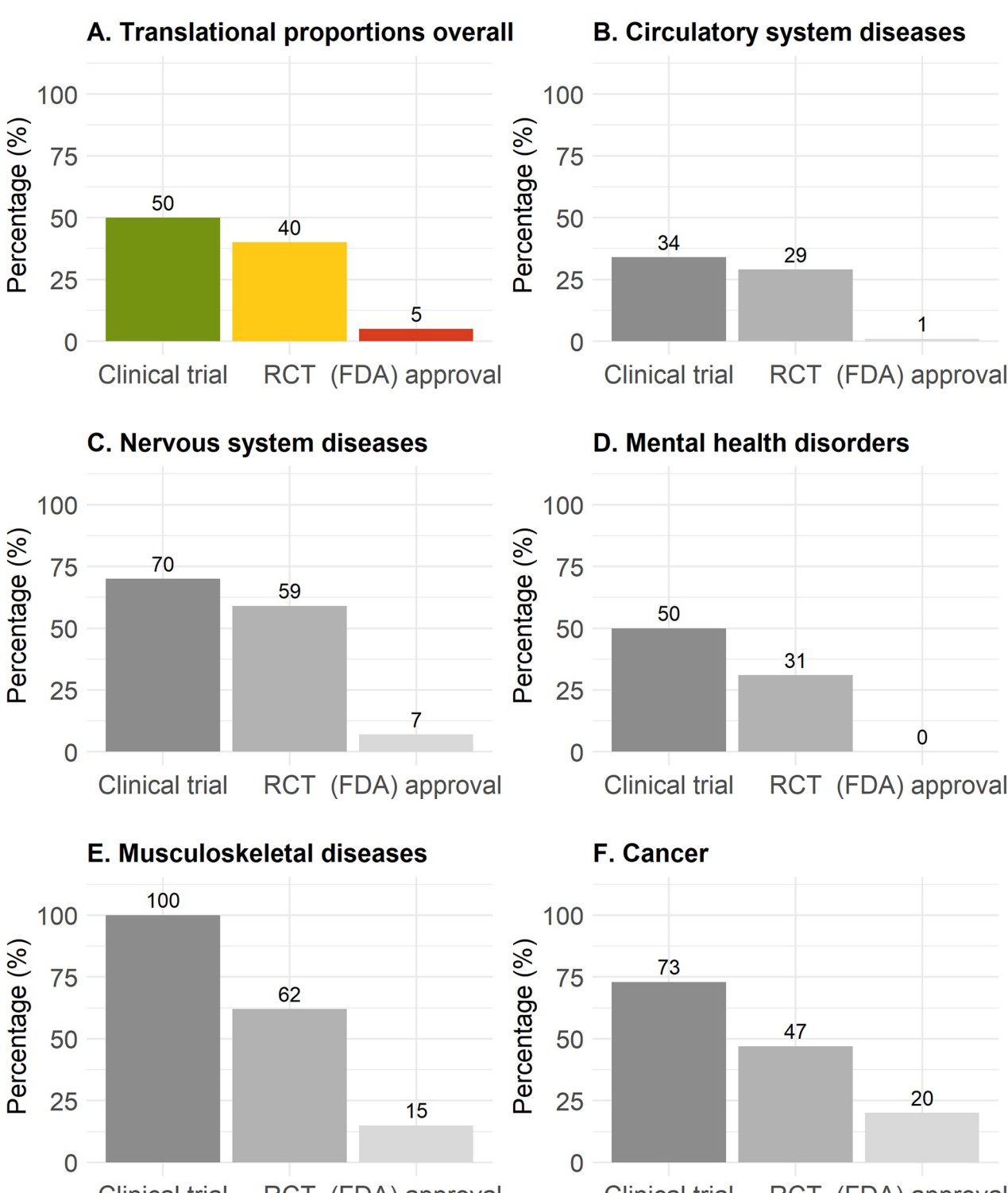

**Fig 3. Translational proportions for all therapies with ≥10 years development time since the first published animal study.** Proportions of translation from animals to any clinical study (green), to an RCT (yellow), or to (FDA) approval (red) overall (A), for circulatory system diseases (B), for neurological diseases (C), for mental health disorders (D), for musculoskeletal diseases (E), and for cancer (F). The data underlying this figure can be found on https://osf.io/frjm4 (Sheet: *Translation*). The code underlying this figure can be found in S1 Code (https://osf.io/9fgru). RCT, randomized controlled trial.

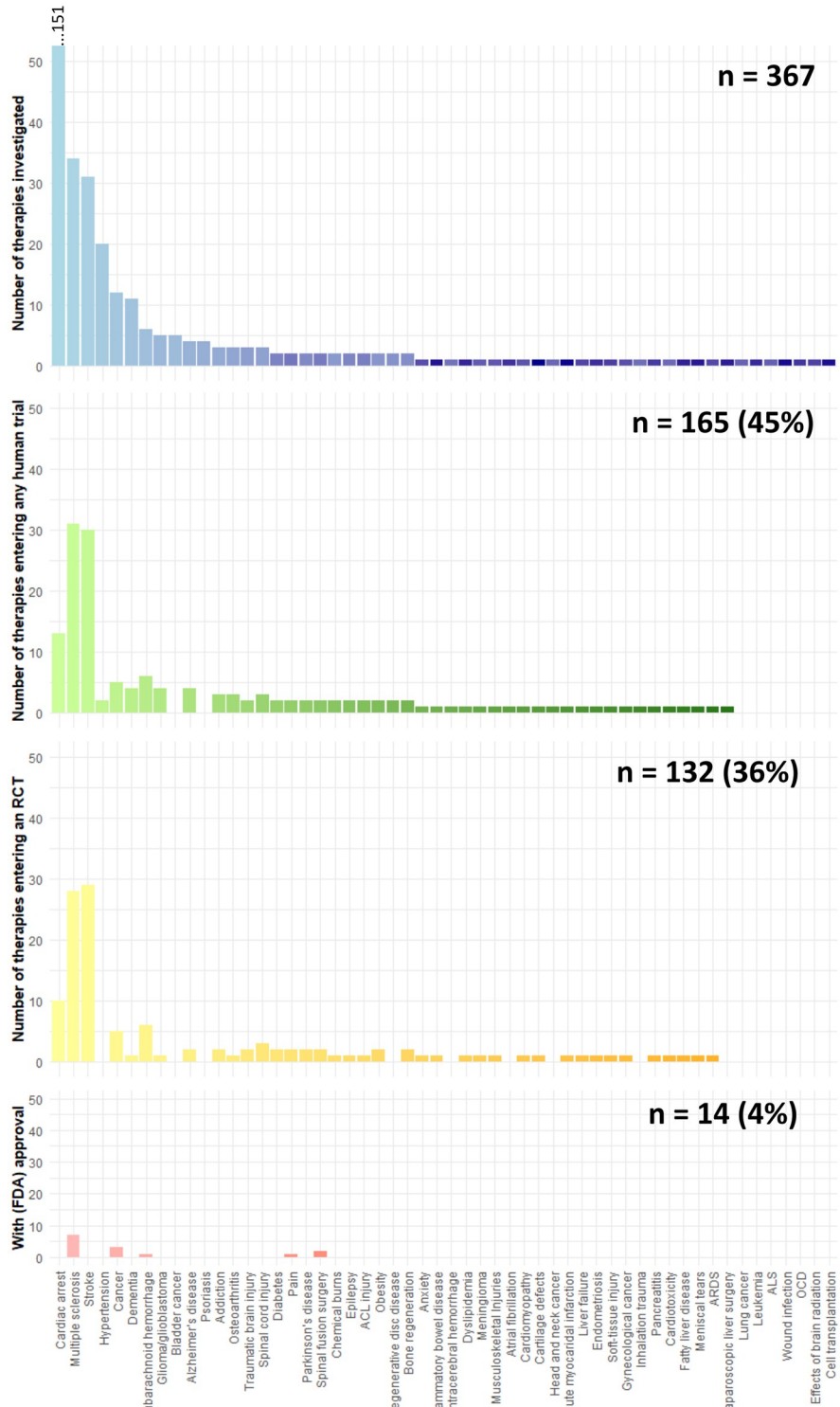

**Fig 4. Progression of therapies through different stages of clinical development, categorized by disease or condition.** The quantity of therapies that are initially tested in animal studies (blue), subsequently entering any clinical trial (green), advance to a randomized controlled trial (yellow), and eventually achieve regulatory approval (red). The total number of therapies for each category is annotated at the upper right corner of the respective graph. The data underlying this figure can be found on https://osf.io/frjm4 (Sheet: *Diseases*). The code underlying this figure can be found S1 Code (https://osf.io/9fgru). ACL, anterior cruciate ligament; ALS, amyotrophic lateral sclerosis; ARDS, acute respiratory distress syndrome; OCD, obsessive compulsive disorder.

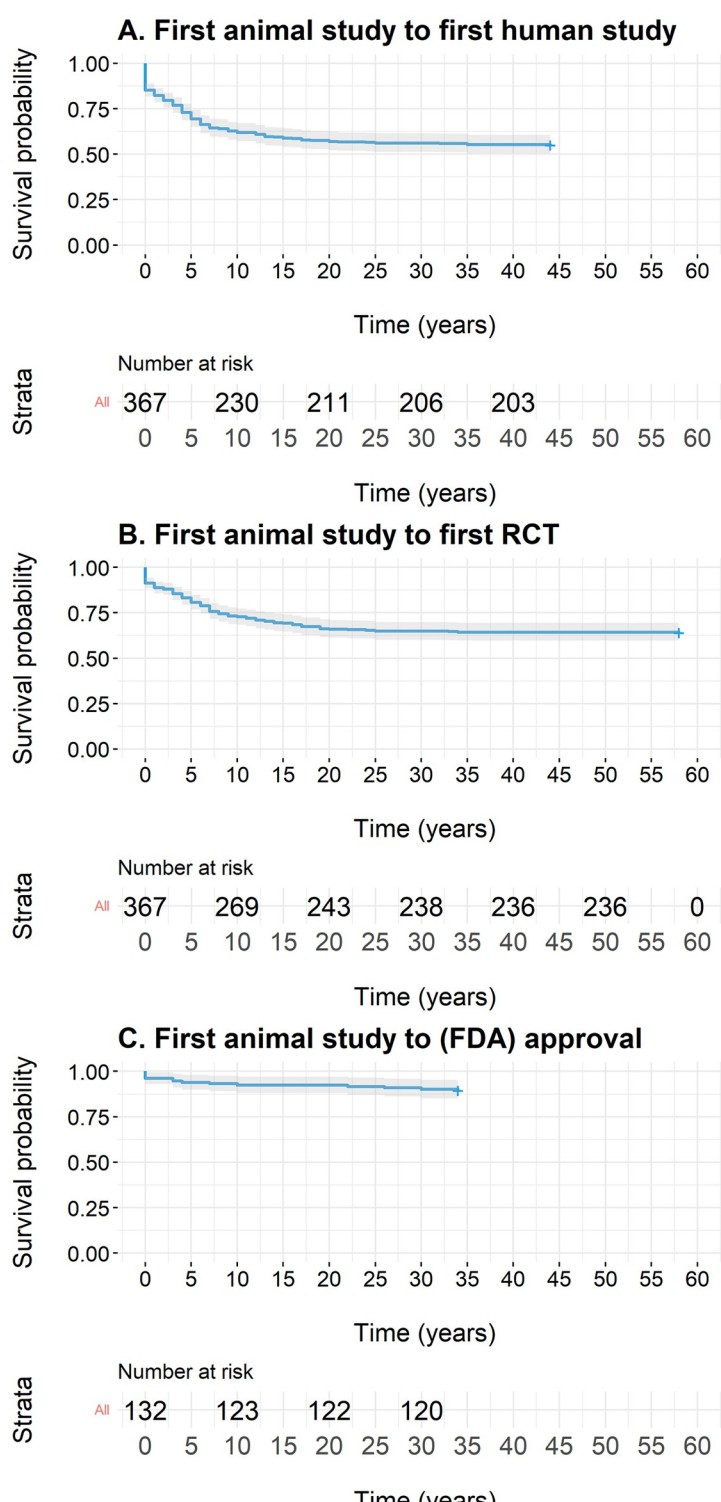

**Fig 5. Lag times for clinical therapy development from first animal study.** Lag times for therapies from first animal study to any clinical study (A), to a randomized controlled trial (RCT, B), or to (FDA) approval (C). The data underlying this figure can be found on https://osf.io/frjm4 (Sheet: *Translation*). The code underlying this figure can be found S1 Code (https://osf.io/9fgru). RCT, randomized controlled trial.

**Table 2. Outcome of animal studies, clinical studies, and RCTs.**

|  | Positive (%) | Neutral (%) | Mixed (%) | Negative (%) | Total |
|---|---|---|---|---|---|
| **Animal studies** | 1,181 (79%) | 278 (19%) | 10 (<1%) | 27 (<2%) | 1,496 |
| **Clinical studies** | 317 (62%) | 178 (35%) | 5 (<1%) | 15 (<3%) | 515 |
| **RCTs** | 111 (50%) | 103 (47%) | 2 (<1%) | 4 (2%) | 220 |

We restricted our analysis to therapies that were the subject of 5 or more published animal studies, encompassing 62 therapeutic interventions.

The data underlying this table can be found on https://osf.io/frjm4 (Sheet: *Mastersheet_including_RoB*).

RCT, randomized controlled trial.

Notably, in several instances, the first animal experiment was documented after the first clinical trial (49 therapies, representing 31%) or RCT (28 therapies, accounting for 22%).

## Concordance between animal and human studies

In evaluating concordance, we restricted our analysis to therapies that were the subject of 5 or more published animal studies, encompassing 62 therapies. These therapies were examined across 1,496 animal studies, 515 clinical studies, and 220 RCTs. Out of these, positive outcomes were identified in 1,181 (79%) animal studies, 317 (61%) clinical studies, and 111 (50%) RCTs (Table 2). The overall alignment between positive outcomes in animal and clinical studies, represented by the relative risk, was 0.86 [95% CI: 0.80 to 0.92] (Fig 6). This alignment was especially pronounced in therapies for neurological diseases at 0.99 [0.66 to 1.06], involving 23 therapies, and circulatory system diseases (inclusive of stroke) at 0.88 [0.76 to 1.01], involving 12 therapies. Conversely, lower concordance was observed for digestive system diseases (0.86 [0.75 to 0.99], 7 therapies), musculoskeletal diseases (0.67 [0.515 to 0.862], 6 therapies), cancer (0.66 [0.49 to 0.88], 3 therapies), and mental health disorders (0.60 [0.37 to 0.97], 7 therapies) (S2–S8 Figs).

## Discussion

### Main findings

Our umbrella systematic review evaluated (1) the proportion of therapies which translate from animal studies to human application, including timeframes; and (2) the consistency between animal and human study results. We observe a notable consistency between results from animal and human studies including a relatively large proportion of therapeutic interventions entering early clinical trials. However, only a minority of therapeutic intervention achieved regulatory approval.

### Findings in the context of existing evidence

**Translation across biomedical fields.** Our review shows a high consistency between findings from animal and human studies, similar to studies outside of therapy translation [44]. In addition, a surprisingly high proportions of therapies entered early clinical development: half of the therapeutic interventions made the transition from animal studies to early human clinical studies (34% to 100% across different biomedical fields). Furthermore, 40% of these therapies progressed to the more rigorous RCT stage (29% to 62% for different biomedical fields). However, a strikingly low proportion—only 5%—of therapies achieved official approval (0% to 20% across the biomedical spectrum).

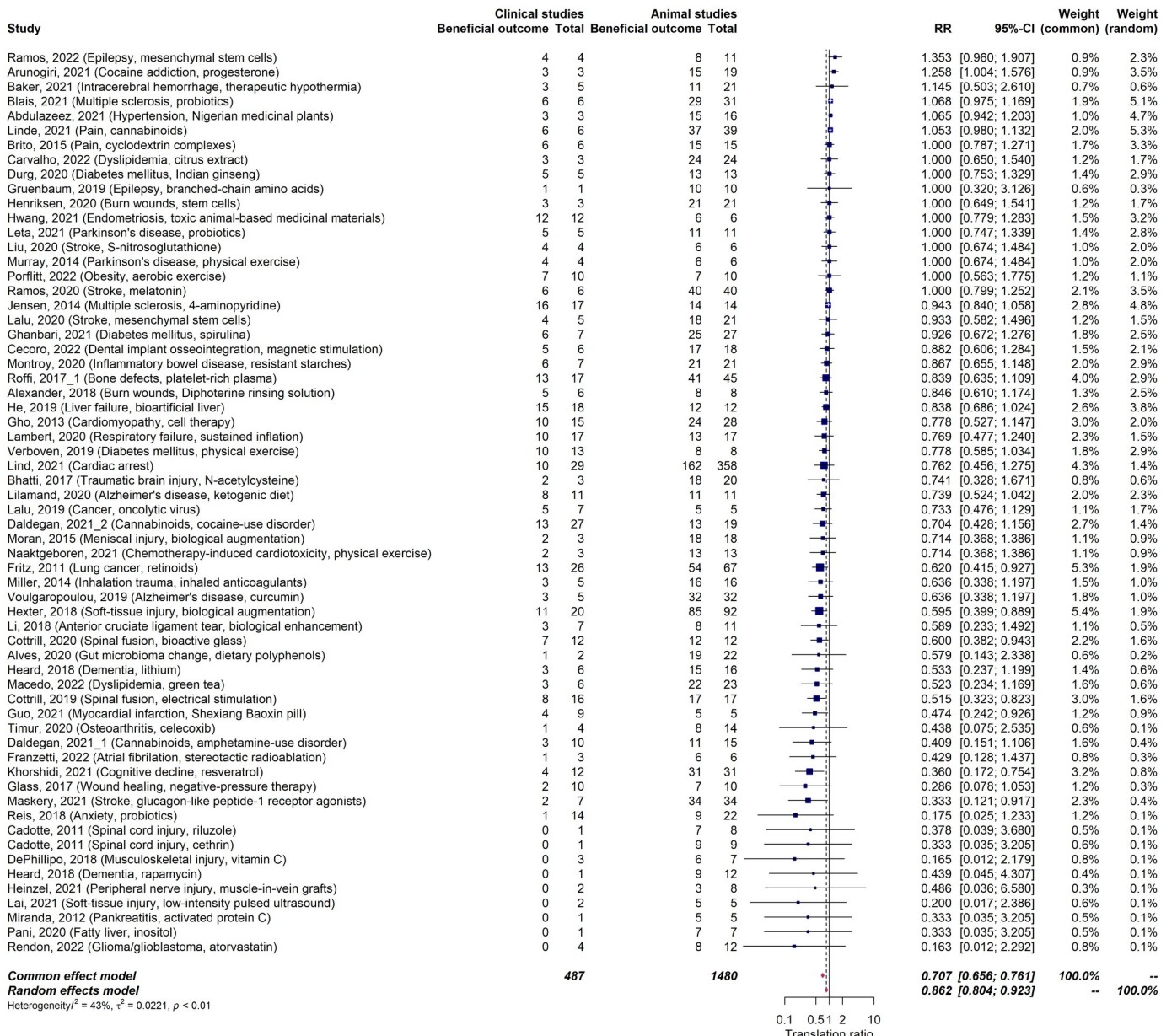

**Fig 6. Meta-analysis on concordance rate between animal and human studies (relative risk).** Forest plot of concordance rates (relative risks) between animal and human studies. A random-effects model was fitted to the data. The data underlying this figure can be found on https://osf.io/frjm4 (Sheet: *Mastersheet_including_RoB*). The code underlying this figure can be found S1 Code (https://osf.io/9fgru). CI, confidence interval; RR, relative risk.

How can we make sense of the fact that animal studies and early clinical trials seem to show promise, yet there is very limited official approval for these therapies? There are 2 possible explanations: One scenario is that the strict requirements of RCTs and regulatory approval are causing many potentially valuable treatments to be left behind. The other scenario is that both animal studies and early clinical trials may have limitations in their design, such as a lack of proper randomization and blinding, which affects their internal validity [45]. This could lead to unreliable findings in both domains, ultimately resulting in the exclusion of these therapies in more rigorous clinical trial settings like RCTs. This line of reasoning also includes the so

called efficacy-effectiveness gap, i.e., the differences in outcomes between patients treated in ideal and controlled circumstances of clinical trials versus real-world scenarios [46,47].

We lean towards the second scenario for 2 reasons. First, as therapies progress to more rigorous study designs, their numbers do decrease as shown by our data, which contrasts to the mostly small and uncontrolled early clinical trials where these therapies were initially tested. Second, drawing from the field of clinical neurology, many therapies that have shown promise in animal studies and early trials reported as successful candidates herein, such as melatonin and mesenchymal stem cells for stroke, have not yet become standard clinical practice [48]. A similar pattern can be seen in other neurological diseases like Alzheimer's disease and spinal cord injury, where there are several therapies with promising preclinical results but limited practical translation [49,50].

**Potential hurdles for successful translation.**   Several factors contribute to the challenges in translating therapies from animal models to human application, as discussed by many of the included reviews. First, there is a notable discrepancy in the contexts of animal testing versus human application. For example, treatment strategies tested on young, healthy animals, such as those for stroke, may not directly apply to the more complex scenarios of elderly patients with multiple health conditions. Second, there is an overall poor quality of many animal studies. These studies often have inherent design flaws, lacking critical elements like blinding or randomization. This absence can bias the results and affect their applicability to the human case, i.e., their external validity. Third, there seems to be a disconnect between animal and human research [51]. This warrants a stronger focus on educating a new generation of translational scientists [52]. Fourth, when it comes to human studies, they can suffer from being underpowered or relying on outcome measures that do not capture the efficacy of a treatment [53]. For example, early phase clinical trials testing interventions for neurological diseases are commonly underpowered [54]. Similarly, clinical trials may use trial outcomes not genuinely reflecting real-world patient settings such as complex composite outcomes, commonly seen in trials of cardiovascular diseases [55] or assessing cognitive domains in dementia trials not relevant for patients [56]. Lastly, animal and human studies commonly address different questions: whereas animal studies tend to focus on mechanisms, human studies tend to focus on effectiveness of an intervention.

**How can we improve translation?**   While finding a straightforward solution is challenging, we emphasize 2 crucial elements to enhance the transition from preclinical studies to clinical applications: the robustness and generalizability of data. ALS research illustrates data robustness issues, with treatments effective in animal models often failing in human studies [57]. On the other hand, the variability in experimental protocols and tools affects how generalizable the results are [58,59]: drugs effective across diverse laboratory settings tend to promise better outcomes in human studies [60]. In addition, outcomes from animal and early clinical studies must align with actual clinical needs [59]. Incorporating these measures could not only streamline drug development but also positively impact animal welfare by reducing research waste [61,62].

Could low translation be an innate characteristic of translational animal research? Instead of disposing animal research due to low translation, a more telling comparison might be drawing parallels between translational rates in animal research and sectors like medical device approvals, where development largely bypasses animal use [63]. It might emerge that translation proportions could be similarly modest in these animal-free sectors.

We did not systematically assess differences in animal models or therapy parameters such as therapy type (drug versus non-drug) for different disease categories—for example, comparing conditions where animal and human therapy outcomes show high consistency (such as neurological or circulatory system diseases) against those with lower consistency (such as

mental health issues or cancer). Future research could explore these factors as potential moderators on the alignment of results between animal studies and human applications. Additionally, future studies should examine how the source of trial sponsorship, comparing investigator-initiated versus industry-sponsored trials, influences translational success, given that industry-sponsored trials appear to exert a greater influence on clinical guidelines [64].

## Limitations

The findings of this study come with notable limitations:

First and most importantly, the therapies herein included were identified through systematic reviews specifically focusing on translation which will likely bias the factual translation. Such reviews may be more commonly conducted in fields where at least some therapies have already achieved clinical translation.

Second, the journey of transitioning a therapy from animal experiments to human application is intricate, bearing a multidimensional nature [43]. Our methodology, however, simplified this process by categorizing clinical studies as positive, neutral, or negative.

Third, the outcomes of clinical studies were classified based on the authors' conclusions, which may lead to biased interpretations if the authors frame their findings to support a beneficial therapeutic outcome [65]. This phenomenon, known as "spin," could result in an overestimation of clinical trials with beneficial outcomes.

Fourth, it is prudent to recognize that animal studies can have indirect benefits in the translation process. For example, they might enhance our mechanistic comprehension of diseases, even if not directly leading to a successful therapeutic application in humans.

Fifth, this umbrella review is confined to studies published in English, which may lead to the omission of relevant data published in other languages.

## Strengths

Our umbrella review also has strengths:

First, we employed a rigorous systematic review methodology, which aids in mitigating potential biases.

Second, it offers a comprehensive perspective by spanning a variety of biomedical fields, therapeutic modalities, and human diseases, and including a relatively large number of therapies and diseases.

Third, we have undertaken both qualitative and quantitative assessments of translational rates, constituting sensitivity analyses.

## Conclusions

Our umbrella review presents translational proportions across various biomedical fields, detailing the progression times from animal studies to clinical development. Although the consistency between animal and early clinical studies was high, only a minority of therapeutic interventions achieved regulatory approval. To enhance development of therapies for clinical application, it is imperative to emphasize the robustness and generalizability of experimental approaches, ensuring rigorous animal and human research.

## Supporting information

**S1 Data. Supplementary data.**
(PDF)

**S2 Data. Supplementary data.**
(DOCX)

**S1 Fig. Translational proportions for all therapies (neglecting development time).**
(DOCX)

**S2 Fig. Lag times for clinical therapy development from first animal study only considering therapies which transitioned to a clinical trial.**
(DOCX)

**S3 Fig. Meta-analysis on concordance rate for neurological diseases (relative risk).**
(DOCX)

**S4 Fig. Meta-analysis on concordance rate for circulatory system diseases (relative risk).**
(DOCX)

**S5 Fig. Meta-analysis on concordance rate for diseases of the digestive system (relative risk).**
(DOCX)

**S6 Fig. Meta-analysis on concordance rate for musculoskeletal diseases (relative risk).**
(DOCX)

**S7 Fig. Meta-analysis on concordance rate for cancer (relative risk).**
(DOCX)

**S8 Fig. Meta-analysis on concordance rate for mental health disorders (relative risk).**
(DOCX)

**S1 Table. Studies, interventions, and development status of included systematic reviews/ therapeutic interventions.**
(DOCX)

**S2 Table. Translational assessment of interventions for neurological diseases.**
(DOCX)

**S3 Table. Translational assessment of interventions for diseases of the musculoskeletal system and connective tissue.**
(DOCX)

**S4 Table. Translational assessment of interventions for psychiatric disorders.**
(DOCX)

**S5 Table. Translational assessment of interventions for diseases of the circulatory system.**
(DOCX)

**S6 Table. Translational assessment of interventions for diseases of the digestive system.**
(DOCX)

**S7 Table. Translational assessment of interventions for cancer.**
(DOCX)

**S8 Table. Translational assessment of interventions for other disorders/conditions.**
(DOCX)

**S9 Table. Number of therapeutic interventions tested in animals, entering any clinical trial, any RCT, and eventually obtaining regulatory approval per disease/condition**

**(n = 54).**
(DOCX)

**S1 Code. Analysis code.**
(R)

## Acknowledgments

We thank Emma-Lotta Säätelä and Nik Bärtsch for help with data curation.

## Author Contributions

**Conceptualization:** Benjamin V. Ineichen, Eva Furrer, Servan L. Grüninger, Malcolm R. Macleod.

**Data curation:** Benjamin V. Ineichen, Eva Furrer, Servan L. Grüninger, Wolfgang E. Zürrer.

**Formal analysis:** Benjamin V. Ineichen.

**Methodology:** Benjamin V. Ineichen, Eva Furrer, Servan L. Grüninger, Malcolm R. Macleod.

**Supervision:** Malcolm R. Macleod.

**Visualization:** Benjamin V. Ineichen.

**Writing – original draft:** Benjamin V. Ineichen, Malcolm R. Macleod.

**Writing – review & editing:** Eva Furrer, Servan L. Grüninger, Wolfgang E. Zürrer.

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
