## [Editor Report · Decision Letter 0]

11 Dec 2023

Dear Dr Ineichen, 

Thank you for submitting your manuscript entitled "From mice to men: Evaluating the success rate of biomedical therapies' journey from animals to humans – an umbrella review" for consideration as a Meta-Research Article by PLOS Biology. Please accept my apologies for the delay incurred during our annual team retreat.

Your manuscript has now been evaluated by the PLOS Biology editorial staff, as well as by an academic editor with relevant expertise, and I'm writing to let you know that we would like to send your submission out for external peer review.

Once your full submission is complete, your paper will undergo a series of checks in preparation for peer review. After your manuscript has passed the checks it will be sent out for review. To provide the metadata for your submission, please Login to Editorial Manager (https://www.editorialmanager.com/pbiology) within two working days, i.e. by Dec 13 2023 11:59PM.

Kind regards,

Roli Roberts

Roland Roberts, PhD

Senior Editor

PLOS Biology

rroberts@plos.org

---

## [Decision Letter · Decision Letter 1]

20 Feb 2024

Dear Dr Ineichen,

Thank you for your patience while your manuscript "From mice to men: Evaluating the success rate of biomedical therapies' journey from animals to humans – an umbrella review" went through peer-review at PLOS Biology. Your manuscript has now been evaluated by the PLOS Biology editors, an Academic Editor with relevant expertise, and by three independent reviewers.

You'll see that reviewer #1 is very positive and has a list of minor textual and presentational requests. Reviewer #2 is also positive and has no requests. Reviewer #3 is also favourable, and has a number of requests, most of which relate to clarity around the methodology. However, while we appreciate that this reviewer's final request, which is to update the study before resubmitting, will further improve the paper, we will leave it to your discretion.

IMPORTANT: please also attend to the following:

a) Please comply with PLOS' Data Policy (https://journals.plos.org/plosbiology/s/data-availability); specifically, we need you to supply the numerical values underlying Figs 2. 4ABC, 5, S2ABC, S3, S4, S5, S6, S7, S8, either as a supplementary data file or as a permanent DOI’d deposition. I note that your study is already preregistered with OSF (https://osf.io/jh2d8), so perhaps you could deposit it there?

b) Please cite the location of the data clearly in all relevant main and supplementary Figure legends, e.g. “The data underlying this Figure can be found in S1 Data” or “The data underlying this Figure can be found in https://doi.org/10.5281/zenodo.XXXXX”

c) Please make any custom code available, either as a supplementary file or as part of your data deposition.

d) Your Data Availability Statement currently says “All data and code that support the findings of this study are available from the corresponding author, BVI, upon reasonable request, at benjamin.ineichen@uzh.ch” - this is not compliant with PLOS Data Policy, so please comply and update (see above).

e) Your Abstract is currently structured; please change it to an unstructured format (https://journals.plos.org/plosbiology/s/submission-guidelines#loc-abstract).

f) Please remove the beginning and end of your Title, to make it simply "Evaluating the success rate of biomedical therapies' journey from animals to humans"

In light of the reviews, which you will find at the end of this email, we are pleased to offer you the opportunity to address the comments from the reviewers in a revision that we anticipate should not take you very long. We will then assess your revised manuscript and your response to the reviewers' comments with our Academic Editor aiming to avoid further rounds of peer-review, although might need to consult with the reviewers, depending on the nature of the revisions.

**IMPORTANT - SUBMITTING YOUR REVISION**

*Resubmission Checklist*

*Published Peer Review*

*PLOS Data Policy*

*Blot and Gel Data Policy*

Sincerely,

Roli Roberts

Roland Roberts, PhD

Senior Editor

PLOS Biology

rroberts@plos.org

REVIEWERS' COMMENTS:

Reviewer #1:

[identifies himself as Jan-Bas Prins]

This manuscript describes a systematic review of published systematic reviews or scoping reviews with or without meta-analysis which investigate translation of interventions in animal models of human diseases. The authors name their approach an umbrella review. The study adds a dimension to the ongoing debate about the validity and translatability of pre-clinical testing in drug development for humans. Focussing on three phases in the trajectory of drug testing: the pre-clinical phase, the randomised clinical trial phase and approval by the competent authority, in this case the FDA. The authors used published search strategies and a purposely developed one to mine Medline, Embase, and Web of Science. They performed a risk of bias analysis in which they identified several studies with some concerns others with high risk of bias. They started with 3343 records for the first screening and ended with 122 studies for the qualitative synthesis and 62 for the quantitative synthesis/ meta-analysis. The study shows that although animal studies appear to translate better to humans than previously thought, there is still a very low rate of final approval. The manuscript is well-written and demonstrates sound research design, execution, and reporting.

I have only a few remarks and comments to make in order of appearance in the manuscript. 

1. Material and methods: 

a. Lines 74-76 - here it reads that Medline was searched with Ovid Medline interface. In other parts of the manuscript, reference is made to Pubmed, which is an interface to search Medline as well as additional biomedical content. 

b. Line 79 - typo 'SYRLCE'

c. Line 81-84 - a 'broad' and a 'narrow' search strategy were tested for sensitivity before selecting one to the two. A threshold was set at 5% additional studies retrieved with the broad strategy. The authors 'did not'. It would be nice to know what the actual percentage was. 

d. Line 96-97 - non-systematic reviews were excluded but retained for additional references. Only one additional article was found and added to the search results. This low number reflects the quality of the SRs included and could be discussed briefly. 

e. Line 106-107 - add reference (or URL) for Universal Desktop Ruler. 

f. Line 151 - add references (or URLs) to the packages survminer and survival respectively. 

2. Results:

a. Lines 181-185 - It is unclear if studies that were labelled as at high risk of bias or had concerns raised were still included in the analysis and if so, what the potential impact is. 

b. Lines 209-210 - the authors note 'that many animal studies were conducted subsequent to the publication of a corresponding clinical study'. Without the studies at hand, this can be interpreted in different ways. It would be helpful if the authors could elaborate on this notion by adding information to what extent the research questions of the animal studies were overlapping with or novel with respect to the clinical study. 

c. Line 217 - (281 therapies, Figure 3) add A to figure 3. 

d. Line 230 - reference to Supplementary figure 2 - in this figure the term 'survival time' is used. Could the alternative 'lag time' be considered? 

e. Line 235-236 - what is the justification for the cut-off of less than 5 published animal studies?

f. Lines 236-239 - The numbers in Table 2 are different from those in the text. The legend of table 2 should include information on the cut-off number of published animal studies. 

3. Discussion:

a. Lines 243-246 - although it may be considered as beyond the scope of the manuscript, the paper would gain from discussion on the differences between the therapies (and/ or models used) for neurological diseases and circulatory system diseases versus those for diseases with lower concordance. Also, in relation to the section of the discussion (lines 273-280) where the example of field of clinical neurology is presented to demonstrate the limited official approval of therapies while the results of animal studies and early clinical trials were promising. 

b. Lines 291-292 - the fourth potential hurdle for successful translation is the quality of human studies i.e. underpowered and inappropriate outcome measures. These refer to serious flaws in experimental design and outcomes of assessment of protocols by competent authorities. Could the authors add examples or provide some level of quantification? 

c. Lines 312-314 - I am not sure I understand this limitation since only one publication was added to those already identified by the SRs based on the excluded papers.

d. Lines 318-320 - Is this a limitation?

Reviewer #2:

[identifies himself as Jens Minnerup]

This is an excellent review article by leading experts in the field. The article will be of interest to a wide readership. 

Congratulations to the authors. 

I have no further comments.

Reviewer #3:

Thank you for the opportunity to review this very well-written paper evaluating the success rate of biomedical therapies' journey from animals to humans conducted as an umbrella review. This is a very important and timely review. The results are interesting and provide new evidence to inform effective and efficient translational pathways across multiple diseases. 

Comments:

- Quality assessment: Can you please clarify the process for developing these questions and any piloting completed to ensure the clarity of interpretation. Were any inter-rater statistics completed? Are additional details for how to apply these questions able to provided in a supplemental to allow someone else to use in the future or reproduce your scores?

- It reads as though you found reviews and then did some linking of other studies based on interventions identified. Are you able to clarify these processes in text. 

- Line 189: Qualitative summary. There is one subsection focused on translation in different biomedical fields. However, are there other subsections? If not, I would remove the subheading. 

- Line 267: Consider the role of sample size and efficacy (ideal and controlled circumstances) vs effectiveness (real world scenarios). Many interventions have been found to show promise at early clinical trial phase, but the effect is diluted when conducted on a much larger (roe generalisable) sample.

- Can you comment on the leadership of trials ie investigator initiated or industry sponsored? Does this have any role to play in the quantitative outcomes examined?

- Limitations section: acknowledge this review is of English only studies 

- Table 1: As well as the summary numbers, can you please list number of studies per unique disease, and include the number in any study, RCT and FDA per unique disease as well? I know the there is a lot of information in the supplemental, but it would be great if there is a way to bring any summary level (consider using advanced data visualisation options) to the main paper. 

- Table 2: Positive/Negative/Neutral - please include information about this in the methods including which outcome was used to make this determination eg primary or any outcome. 

- Line 96: add (3) in front of at full text … 

- Line 190: consider breaking up the paragraphs for stroke, cardiac arrest and cancer for ease of reading. This may mean expanding a little for some areas. 

- Recommend updating the search prior to re-submission to ensure recency, however, also appreciate the large volume of work that has gone into pulling this review together, and note the findings may not change given the nature of this review. I am happy to be guided by editorial preference and authorship team perspectives here. 

I really enjoyed reading this paper and look forward to citing it in the future! Congratulations on comprehensively pulling this large body of evidence together.

---

## [Editor Report · Decision Letter 2]

3 May 2024

Dear Dr Ineichen,

Thank you for your patience while we considered your revised manuscript "Evaluating the success rate of biomedical therapies' journey from animals to humans" for publication as a Research Article at PLOS Biology. This revised version of your manuscript has been evaluated by the PLOS Biology editors and the Academic Editor. Please accept my apologies for the delay incurred. I should clarify that we had extreme difficulty communicating with the previous Academic Editor, and we have therefore switched to a new Academic Editor who will handle the paper for the rest of its journey.

Based on our Academic Editor's assessment of your revision, we are likely to accept this manuscript for publication, provided you satisfactorily address the points raised by the Academic Editor and the other data and other policy-related requests.

IMPORTANT - please attend to the following:

a) Please address the concerns from the (new) Academic Editor. You can find their actual comments at the foot of this email. My understanding is that they want you to 1. make it clear in the manuscript exactly how the intervention effect was classified, and 2. if you used one of the more problematic methods (e.g. relying on author conclusion), then this should be identified clearly as a limitation, with some discussion of the potential issues (spin, etc.).

b) Please could you change your title to something that encapsulates some of the findings? We suggest something like "Evaluating the success rate of biomedical therapies from animals to humans shows a 5% regulatory approval rate after 10 years"

We expect to receive your revised manuscript within two weeks. 

*Published Peer Review History*

*Press*

Sincerely,

Roli Roberts

Roland Roberts, PhD

Senior Editor

rroberts@plos.org

PLOS Biology

DATA NOT SHOWN?

COMMENTS FROM THE ACADEMIC EDITOR [lightly edited]:

I read the answers to reviewers and the last version of the manuscript. I agree that the authors adequately answers all comments and revised their paper accordingly.

I must nevertheless say that I do not feel very comfortable in their classification of each study as whether the intervention had a positive, negative, mixed, or neutral effect on the corresponding disease outcome.

It seems quite unclear to me how they could do this: which outcome they considered (they state the main study outcome but often its difficult to identify or there are many and we don't know which one was prespecified), did they focus on p-value, effect estimates, confidence interval, consistency with other outcome assessed or did they use the author conclusion, which raises the issue of spin. This is slightly highlighted by reviewer 3 considering a specific study.

This approach is also questionable as the review authors "observed a progressive reduction in efficacy from animal research to early clinical trials and then to phase 3 clinical trials. "

Otherwise the rest of the paper did not raise any issue from me.

---

## [Editor Report · Decision Letter 3]

7 May 2024

Dear Dr Ineichen,

Thank you for the submission of your revised Meta-Research Article "Analysis of animal-to-human translation reveals that only 5% of animal-tested therapeutic interventions obtain regulatory approval for human applications" for publication in PLOS Biology. On behalf of my colleagues and the Academic Editor, Isabelle Boutron, I'm pleased to say that we can in principle accept your manuscript for publication, provided you address any remaining formatting and reporting issues. These will be detailed in an email you should receive within 2-3 business days from our colleagues in the journal operations team; no action is required from you until then. Please note that we will not be able to formally accept your manuscript and schedule it for publication until you have completed any requested changes.

IMPORTANT: You'll see that I've taken the liberty of doing a minor edit to your title (my Editor-in-Chief has an aversion to punctuation in titles).

Sincerely,

Roli Roberts

Senior Editor

PLOS Biology

rroberts@plos.org